# Raman Spectroscopy on Brain Disorders: Transition from Fundamental Research to Clinical Applications

**DOI:** 10.3390/bios13010027

**Published:** 2022-12-26

**Authors:** Jeewan C. Ranasinghe, Ziyang Wang, Shengxi Huang

**Affiliations:** Department of Electrical and Computer Engineering, Rice University, Houston, TX 77005, USA

**Keywords:** Raman spectroscopy, brain disorders, clinical treatment, biomarker identification, statistical analysis

## Abstract

Brain disorders such as brain tumors and neurodegenerative diseases (NDs) are accompanied by chemical alterations in the tissues. Early diagnosis of these diseases will provide key benefits for patients and opportunities for preventive treatments. To detect these sophisticated diseases, various imaging modalities have been developed such as computed tomography (CT), magnetic resonance imaging (MRI), and positron emission tomography (PET). However, they provide inadequate molecule-specific information. In comparison, Raman spectroscopy (RS) is an analytical tool that provides rich information about molecular fingerprints. It is also inexpensive and rapid compared to CT, MRI, and PET. While intrinsic RS suffers from low yield, in recent years, through the adoption of Raman enhancement technologies and advanced data analysis approaches, RS has undergone significant advancements in its ability to probe biological tissues, including the brain. This review discusses recent clinical and biomedical applications of RS and related techniques applicable to brain tumors and NDs.

## 1. Introduction

The human brain is without doubt one of the most fascinating works of nature. It is the central organ of the nervous system that controls the essential activities of humans. Brain disorders arise due to atypical features in brain functional, structural, and biochemical levels. Many of these diseases, including Alzheimer’s disease (AD), Parkinson’s disease (PD), Huntington’s disease (HD), and brain tumors represent public health challenges, as they can have a profound and even debilitating impact on a patient’s life [1,2,3,4]. The level of care required for patients with brain anomalies adds further economic and social burden, highlighting the importance of developing efficient treatments. As a result, the detection of these devastating disorders at an early stage represents paramount importance in precision medicine. Current diagnosis methods are highly reliant on CT, MRI, and PET [5,6,7]. While these techniques offer deep tissue imaging capabilities, they suffer from major drawbacks such as high cost, poor spatial resolution, limited insight into specific molecular information, and the adverse effects of using ionizing radiation. Therefore, developing fast, non-invasive, and cost-effective tools remains a central theme in clinical applications.

Numerous developments in laser spectroscopy have enabled significant progression of vibrational spectroscopy in biological applications. As one special method of various vibrational spectroscopic techniques, RS has been established as the front runner in the clinical diagnosis of brain anomalies [8,9]. It is widely accepted as a noninvasive modality that can provide a wealth of information on the cellular and molecular level due to the inelastic scattering of incident light. Typically, in RS, light from a monochromatic laser interacts with the sample’s vibrational modes, resulting in inelastic photon scattering. These photons are shifted in energy to values different than that of excitation. This is measured as Raman shift and gives information that is specific to chemical bonds. The resulting spectra provide unique information where the Raman shift value provides information about different molecular species and their relative concentration can be evaluated based on the strength of different peaks. Thus, RS can provide a vibrational “fingerprint” of the sample under investigation. More importantly, RS has significant potential in the diagnosis, progression, and evaluation of treatments for brain disorders [10,11]. This is mainly due to its ability to differentiate healthy and diseased tissues that can reveal specific biomarkers based on the stage of the disease. Moreover, RS does not require labeling for detection. Sometimes the signal generated from biological tissues in spontaneous RS is relatively weak. In such situations, surface-enhanced Raman scattering (SERS) is useful in gathering meaningful information [9]. Additionally, this technique can be used in vivo due to the advancement of fiber-optic probes coupled with portable Raman systems. These advances in RS in clinical applications have been further augmented by rapid progress in chemometrics and machine learning (ML) algorithms [5]. Several data analysis methods and ML models such as principal component analysis (PCA), classical least square fitting (CLS), partial least square (PLS), and linear discriminant analysis (LDA) allow for the extraction of hidden information that cannot be accessed through human inspection and basic statistical methods.

In this review, we attempt to shed further light on significant advances and state-of-the-art development of RS in clinical applications of brain disorders. We first start our discussion with the principle of Raman scattering and general spectrometer setup. Then, we discuss a range of Raman techniques such as resonance Raman spectroscopy (RRS), SERS, and variations of RS. After that, we include a brief discussion on statistical analysis tools including ML on Raman spectra as effective tools for biomarker identification of brain disorders. Then, we present different brain disorders categorized under NDs and tumors. Finally, we discuss the challenges and prospects of RS for clinical applications of brain anomalies. We believe that this comprehensive review will stimulate and trigger the understanding of RS as a potential tool in the diagnosis of brain disorders.

## 2. The Principles of Raman Spectroscopy and Related Techniques

When a photon of light interacts with matter, it can be scattered either elastically or inelastically. RS engages the inelastic scattering of light by matter, which was first described by C.V. Raman early in the 20th century. The Raman effect is observed through Stokes and anti-Stokes scattering in which the scattered light has either a lower or higher frequency than that of the incident light, respectively. In the biomedical field, Stokes scattering is the most dominant pathway, and the signal is relatively weak. Only 1 in 10 million photons experience Raman scattering. The difference in energy between the incident light and the Raman scattered light is characteristic of the frequency of the vibrational bond that is excited. Additionally, Raman scattering requires a change in polarizability. The spectrum of the scattered photons is represented as the Raman spectrum, and it shows the intensity of the scattered light as a function of the Raman shift. Raman shift values are an identification of the target molecule, which reflects specific chemical bonds and constitutions. Thus, every molecule has a unique spectrum that can be identified as a vibrational fingerprint allowing for the identification of biological materials such as proteins, lipids, and DNA.

For the measurements of the scattering signal, RS systems are used, which are composed of a light source, the spectrometer, a filter to block the laser line, and a detector (Figure 1f). Lasers are used to provide monochromatic radiation for the excitation of molecules. A key consideration of the experimental design is the choice of laser wavelength. This can depend on various factors such as resonance conditions of the sample, extent of fluorescence, background signal, the sensitivity of the detectors, and signal-to-noise ratio (SNR). Objective lenses are used to focus the light on the sample and to collect the scattered radiation. The scattered light is then analyzed by a spectrometer coupled to a suitable detector. A set of filters (laser line filter and long pass or notch filter) are used to remove excitation radiation and Rayleigh scattered light. A diffraction grating is used to separate useful radiation into constituent wavelengths and is finally detected by a sensitive detection system, commonly by a charge-coupled device (CCD). Large datasets are often required to apply chemometrics and ML analysis to extract meaningful data. Toward this goal, large sampling areas up to centimeters need to be analyzed with suitable approaches. This is when Raman mapping is particularly useful. One possibility is to move the laser or sample in a predetermined pattern to measure the Raman spectrum at every position. Additional approaches involve expanding the laser focus, laser line, light sheet, and wide-field illumination. The resulting Raman maps contain chemical and structural information coupled with spatial information.

### 2.1. Resonance Raman Spectroscopy (RRS)

Spontaneous Raman scattering is inherently weak, and it requires special conditions to magnify the signal. In RRS, the wavelength of the excitation light is tuned to match the electronic transitions of the sample under investigation (Figure 1a). Such resonance conditions result in the enhancement of the signal that could be undetectable under normal conditions. Additionally, RRS only amplifies Raman scattering from a specific vibrational mode. In the literature, enhancement up to six orders of magnitude was reported [13]. This technique allows for the design of an enhancement mechanism without the interference of foreign moieties. One drawback of RRS is increased fluorescence that can interfere with the Raman signal. However, this can be minimized by choosing the proper wavelength for excitation. RRS is becoming a popular tool in identifying NDs and brain cancers. For example, RRS provides information about protein structures and conformations as well as healthy and diseased tissues.

### 2.2. Surface-Enhanced Raman Spectroscopy (SERS)

While conventional RS provides excellent chemical specificity, it is inherently weak. One method for enhancing the weak signal is using metallic substrates to take advantage of the enhanced electric field at the surface of metal nanoparticles caused by localized surface plasmon resonance (LSPR) [14,15,16]. Gold and silver nanoparticles are widely used for SERS experiments, and their properties can be tuned depending on the size, shape, composition, and dielectric environment of the nanoparticle [17]. SERS is a rapid, sensitive, and label-free technique that allows for even single-molecule detection. Therefore, it has clear advantages for diagnostic applications related to NDs. In SERS measurements, the resulting enhancement is maximized when plasmon frequency is in resonance with frequency in incident light (Figure 1a). However, other factors need to be considered, such as nanoparticle clustering and surface adsorption [18].

### 2.3. Other Variations of Raman Spectroscopy

While spontaneous Raman scattering, SERS, and RRS are available as widely researched techniques, other variations of RS are also used in brain diagnosis. Coherent anti-stokes Raman scattering (CARS) and stimulated Raman scattering (SRS) are widely utilized versions of nonlinear RS. The principle behind CARS is to use a pump laser beam and a Stokes laser beam to produce an anti-Stokes signal (Figure 1d). In CARS, an enhanced Raman signal is obtained that is orders of magnitude stronger than spontaneous Raman scattering. SRS is also based on the same principle as CARS to produce resonantly enhanced signals (Figure 1d,e). The amplified Raman signals allow for the label-free detection of target analytes with a high spatial resolution. Tip-enhanced Raman spectroscopy is a combination of SERS and scanning tunneling microscopy (STM). As a result, it has unique advantages such as chemical sensitivity and high spatial resolution. Fiber optic probes and handheld instruments are particularly useful, as they are portable, have small dimensions, and are easy to use in clinical testing.

## 3. Statistical and Machine Learning Analysis for Raman Data

Raman spectra measured on brain samples are high-dimensional, complex, and noisy. To analyze the complicated Raman spectra of brain samples, classical statistical models are frequently used. In diagnosing various brain diseases, PCA is applied to visualize sample patterns and to interpret significant Raman peaks. For example, in studying AD, Fonseca et al. deployed PCA on Raman spectra of mouse brains to visualize the difference between samples with different ages captured by RS (Figure 2a) [19]. Sevgi et al., visualized principal components (PCs) to identify important Raman peaks correlated to PD in Rat brain models (Figure 2b) [20]. Researchers also perform PCA to reduce dimensionality and extract features before further analysis. Huefner et al. applied PCA on Raman spectra and used the generated PCs as inputs to diagnose HD with serum samples [21]. In analyzing molecular processes in brain cancer, Lemoine et al., used 50 PCs produced by PCA and optimized the performances of the classifier (Figure 2c) [22]. Other researchers also reduce the dimension of Raman data with PCA in studying various brain tumors, including gliomas and meningiomas before feeding into the classifiers [23,24,25]. Other statistical tools such as t-distributed stochastic neighbor embedding (t-SNE) can also be used to visualize the differences between Raman spectra of different samples. Wang et al., applied t-SNE to project the high-dimensional Raman spectra of mouse brains with and without AD into two-dimensional plots and visualized the clusters [26]. 

ML is an advanced technique that can recognize patterns and capture minor differences between data clusters. Therefore, it is an excellent tool for analyzing Raman spectra. In recent years, ML methods have been thriving in the clinical diagnosis of brain diseases and the detection of brain cancers. Wang et al., proposed an interpretable ML method with the support vector machine (SVM) and RS to identify potential biomarkers of AD in mouse brains [26]. Specifically, they collected Raman spectra on slices of mouse brains with and without AD, applied SVM to classify AD and non-AD spectra, and identified a spectral feature importance map that reveals the importance of each Raman wavenumber in classifying AD and non-AD spectra (Figure 2d). Desroches et al., also applied SVM to perform in vivo diagnosis of brain cancer with RS [27]. Morais et al., combined PCA and SVM and achieved high performance in differentiating meningioma Grade I and Grade II samples (Figure 2e) [23]. Another popular ML approach is PCA-LDA. Bury et al., identified different tumor statuses with the LDA-PCA approach and achieved high accuracy (Figure 2f) [28]. Other researchers also demonstrated that PCA-LDA is efficient in studying different brain cancers and tumors with Raman spectra [21,22,23,24]. Partial least squares discriminant analysis (PLS-DA) is also widely used in investigating brain cancer. Abramczyk et al., classified Raman spectra of tissue from grade IV medulloblastoma and non-tumors using PLS-DA [29]. Other researchers performed classifications with PLS-DA and achieved high accuracy in classifying different tumoral brain tissues [30,31].

**Figure 2 biosensors-13-00027-f002:**
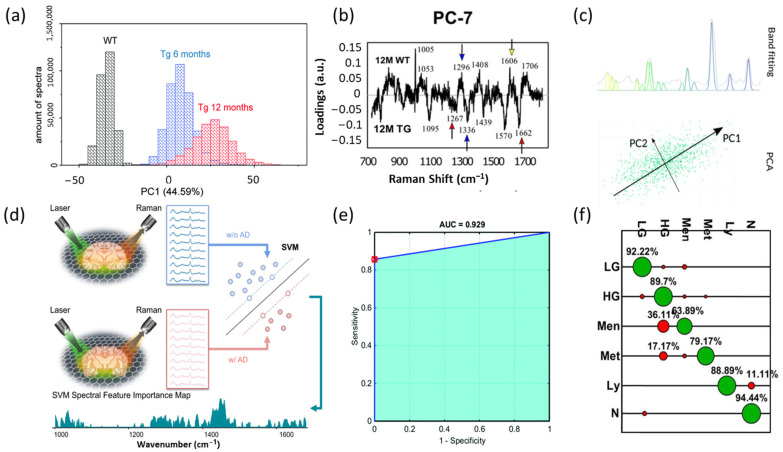
Illustration of applicability of statistical and ML methods on RS in brain clinical applications. (**a**) Histogram showing the differences between groups by PCA of the Raman spectral of brain samples. Twelve-month-old wild-type mice WT (black), six-month-old transgenic mice Tg6 (blue), and twelve-month-old transgenic mice Tg12 (red). Adapted with permission from Reference [19] © 2021 The Royal Society of Chemistry. (**b**) Visualization of the loadings of PC7. Positive side wild-type rat (WT) and negative side transgenic rat (TG) brain samples. Adapted with permission from Reference [20] © 2021 Frontiers. (**c**) Raman spectra were processed by both band fitting and PCA with 50 principal components before being fed into classifiers. Adapted with permission from Reference [22] © 2019 The Royal Society of Chemistry. (**d**) Workflow of Raman signals’ data collection, preprocessing, and ML classification and interpretation to differentiate AD/non-AD Raman spectra of brain samples. Adapted with permission from Reference [26] © 2022 American Chemical Society. (**e**) Receiver operating characteristic (ROC) curve for PCA-QDA. AUC: area under the curve. AUC values between 0.7 and 0.8 are considered acceptable, between 0.8 and 0.9 are considered excellent, and above 0.9 are considered outstanding. Adapted with permission from Reference [24] © 2019 The Royal Society of Chemistry. (**f**) Confusion matrix for PCA-LDC model classifying: non-tumor brain tissue (N); low-grade glioma (LG); high-grade glioma (HG); meningioma (Men); metastasis (Met); lymphoma (Ly). Green is correctly classified, whereas red is incorrectly classified. Adapted with permission from Reference [28] © MDPI 2019.

There are many other ML methods applied on Raman spectra of various tissues and diseases that can be extended to the Raman spectra of brain samples. Perumal et al., applied logistic regression to evaluate the diagnostic biomarker of ovarian cancer [30]. Tree-based ML methods are also used in classifying Raman spectra. In classifying the Raman spectra of the receptor-binding domain of SARS-CoV-2 and MERS-CoV virus, Zhang et al. applied random forest and XGBoost and achieved accuracies over 95% [18]. Ye et al., also applied XGBoost to classify Raman spectra of different strains of SARS-CoV-2 [31]. Convolutional neural networks are also used in classifying Raman spectra of different samples [32,33,34]. For example, Ma et al., achieved an accuracy of 92% with 1D-CNN in diagnosing breast cancer.

ML models are also useful in reducing signal noise and enhancing SNR. Variational autoencoder (VAE) is a convolutional neural network architecture with encoding and decoding stages [35,36,37]. The VAE is trained in an unsupervised way that reconstructs the input spectra. The encoder compresses the input in a latent space. With the lower dimension of the latent space, the noise is removed during the reconstruction by the decoder. He et al., applied the VAE to improve the signal-to-noise ratio of Raman spectra and to significantly increase the accuracy of tumor subtype detection [36].

## 4. Applications of Raman Spectroscopy in Brain Diseases

The seriousness of brain disorders has led to significant investment into research that can identify diagnoses, therapies, and preventive pathways of these deadly diseases. This broad category of brain disorders can vary depending on symptoms and severity. Out of many diseases that can affect the brain, NDs and brain tumors are more prevalent. The major hallmark of NDs is protein accumulation. However, abnormal conformational properties including amyloidosis, tauopathies, α-synucleinopathies, and proteinopathies are also responsible for the development of NDs. RS has great potential in identifying these diseases and was successfully applied in clinical studies. Table 1 compares mechanisms, biomarkers, Raman sensitivity, and diagnose methods other than Raman spectroscopy for NDs. 

The application of Raman techniques to clinical samples and animal models is still at an early stage and still needs close collaborations between spectroscopists, material scientists, biomedical engineers, and clinicians, who are required to make the clinical transformation of RS a reality. However, emerging reports have demonstrated the promising potential of RS in clinical settings [58,59,60,61]. The exploitation of Raman techniques in clinical laboratories is mainly dependent on the availability of portable Raman systems and the advancements in miniaturization [62,63]. Tanwar et al., and Allakhverdiev et al., described the clinical applications of Raman spectroscopy on four different avenues including disease diagnosis, surgical guidance, therapeutic monitoring, and metabolite monitoring [63,64]. The Raman spectrum of biofluids such as urine, saliva, serum, and tears induce many peaks representative of the plethora of cellular constitutes. Additionally, biomarker identification of such biofluids has the potential to study various physiological and pathological processes. Differences between healthy samples and pathologic conditions are exhibited as peak shifting and different intensities of the Raman spectrum. Additionally, there could be an emergence of new peaks allowing for precise identification of pathologies. DePaoli et al., described three main system types required for the exploitation of RS in neurosurgery [62]. First, there should be availability of single-point RS probes for intact tissue assessment. Another requirement is access to portable Raman microscopes for histopathological evaluation after tissue resection. Finally, there should be availability of endoscopic imagers for surgical guidance. In this section, we introduce the state-of-the-art developments of RS in the diagnosis of the above-mentioned brain anomalies.

### 4.1. Neurodegenerative Diseases

NDs constitute chronic, progressive, and irreversible diseases that can affect many of the body’s activities, such as movements, talking, heart function, balance, and breathing. Some of these diseases include AD, PD, HD, and so on. The diagnosis and treatment of NDs represent a significant challenge to healthcare specialists, as most of the symptoms become evident at later stages. Research in NDs is currently undertaken at a rapid pace. Promising results from various studies have led to improving the understanding of risk factors associated with this disease. This includes age, family history, susceptibility genes, lifestyle choices, environmental factors, education, and gender, to name a few. Although the pathogenesis and the degeneration mechanisms are different for each disease, they share common characteristics such as protein misfolding and aggregation, enabling RS as a handy tool in diagnosis. A comprehensive understanding of protein structure is the key to controlling disease progress. RS of healthy tissue is composed of many constituents, and when certain pathologic conditions arise, there will be a change in Raman shift values and intensities of the peaks. This provides a unique pathway to identify biomarkers related to NDs with the aid of chemometrics.

#### 4.1.1. Alzheimer’s Disease (AD)

AD is the most prevalent neurodegenerative disorder associated with weight loss, memory deficits, and cognitive decline. Various studies support the finding that the accumulation of Amyloid-β (Aβ) in the brain is responsible for the progression of AD. Additionally, tau proteins are responsible for the initiating event of AD. RS and related methods have evolved as sensitive methods for biomarker screening of AD.

(a)Fundamental Investigations Related to AD

Conventional RS has been applied to the investigation of biomarkers related to the progression of AD using post-mortem human brain tissues and biofluids. SRS has been applied in studying amyloid plaques in AD. A recent work by Ji et al. reported three-color SRS imaging of amyloid plaques of AD [65]. The researchers were able to distinguish misfolded and normal proteins by measuring the spectral shifts of the amide-I bands (Figure 3a). The results revealed an approximately 10 cm^−1^ blue shift of the amide-I band, which was obvious in both frozen and fresh tissue samples. In another work, Lochocki used the SRS-based multimodal technique to detect amyloid deposits in snap-frozen AD human brain tissue [66]. The SRS results revealed the transformation of protein to a β-sheet structure.

In an early report, Sudworth et al. utilized RS in combination with PCA analysis to discriminate AD disease status in terms of protein conformation changes [67]. A similar study conducted by Hu and coworkers verified for the first time that Raman signatures from the brain hippocampus could help to explore the pathogenesis of AD [68]. They injected Aβ_25–35_ into the hippocampus CA1 regions of rats, and an experimental procedure was carried out using a 785 nm laser for spectral acquisition. To rule out the effects from injection itself, the researchers also carried out continuous monitoring of the detailed spectral changes. Compared with the spectra of normal rats, that of AD rats is characterized by signature peaks and normalized intensity differences. For example, a shoulder Raman peak at 1670 cm^−1^ assigned to the C=O stretching vibration of the β sheet secondary structure has been used to distinguish AD and healthy samples. Additionally, normalized intensities of Raman peaks at 1065, 1088, 1130, 1300, and 1440 cm^−1^ are dominated in AD samples demonstrating hallmarks in the progress of AD, such as Aβ deposition, increase in cholesterol, and increase in slightly hyperphosphorylated tau.

Michael et al. reported the utility of RS as a beneficial technique to analyze eye lens diseases caused by protein aggregation [69,70]. The study showed that protein aggregates of the hippocampus and cortical cataracts of eye lenses have significantly different Raman profiles. More recently, Popp and coworkers carried out biochemical characterization of retinal neurodegeneration of an AD model by ex vivo Raman investigations [71]. Investigated samples captured the layered structure of the retina using a spatial resolution of 2 µm in Raman-based imaging. This finding was further supported by the hematoxylin and eosin (H&E) staining procedure. The layers were identified based on Raman signature peaks attributed to nucleic acids, Rhodopsin, lipids, and proteins (Figure 3b,c). For targeted in vivo applications, with a focus on AD detection, en face Raman imaging was processed, revealing important biochemical information. The study revealed that an early-state biochemical change in the protein composition precedes more conventional late-stage structural changes and pathological pathways of AD. Furthermore, researchers achieved 85.9% accuracy in chemometric analysis. 

Two-dimensional (2D) materials are layered crystalline materials characterized by a list of exotic properties. The family of 2D materials includes graphene, transition metal dichalcogenides (TMDs), hexagonal boron nitride (h-BN), black phosphorous (BP), MXenes, etc., [72,73,74,75]. They are highly applicable in optical bioimaging, therapy, and tissue engineering [76,77]. These fascinating materials have been explored in the landscape of NDs and other brain disorders [57,78,79,80]. Choi et al. demonstrated the reliability of graphene oxide (GO)-hybrid nano arrays as the SERS-sensing substrate for detecting 10^−4^ to 10^−9^ M concentrations of dopamine [53]. Very recently, our group reported that rapid biomarker screening of AD could aid timely and effective treatment measures [25]. Briefly, we utilized graphene-assisted RS and ML techniques to differentiate mouse brains with and without AD. We collected data from three different brain regions: the cortex, hippocampus, and thalamus. In the case of graphene in contact, it exhibited much higher SNR for all three brain regions. We explained the results based on exceptional properties of graphene such as high thermal conductivity and fluorescence quenching capability. More importantly, using ML interpretation, we identified three molecules (triolein, phosphatidylcholine, and actin) that are positively correlated with AD and two molecules (cytochrome and glycogen) that are negatively correlated to AD.

SERS can also help provide molecular fingerprints of brain tissues. Liu and coworkers used black phosphorous and a gold nanoparticle nanohybrid (BP-AuNPs) as a SERS substrate to understand the molecular composition of various encephalic regions [81]. SERS measurements were achieved through Balb/c mice and 785 nm excitation. Spectra were collected from four encephalic regions (cerebral cortex, hippocampus, thalamus, and hypothalamus) showing spectral differences among different regions. The different spectra obtained by subtracting the left and right hemispheres of four encephalic regions provide insight into variation in the local biochemical environments. In another study, Demeritte and colleagues designed core-shell nanoparticles modified with GO for selective separation and label-free identification of AD biomarkers. Here, Aβ was magnetically separated from the whole blood sample and achieved a detection limit of 100 fg/mL [52]. It has been shown that metal ions, especially Fe^3+^, Cu^2+^, and Zn^2+^, have a significant impact on Aβ aggregation [82,83]. The link between metal ions and disordered proteins in NDs is well known and widely studied using SERS. Zhou et al. reported the potential of SERS spectra to reveal real-time imaging of Aβ aggregation under different conditions [84]. In their study, AuNPs synthesized in situ with Aβ monomer and fresh mouse brain were used as a template to understand the role of metal ions on Aβ aggregation. Their results mentioned that Cu^2+^ and Zn^2+^ ions of low concentration promote fibril formation, while Fe^3+^ and Zn^2+^ of high concentration inhibit fibril formation (Figure 3d). Other Raman techniques on biomarker detection of AD and the dynamical behavior of Aβ aggregation are progressing gradually, offering valuable insight for the clinical transformation of RS in disease prevention [85,86,87].

Mild cognitive impairment (MCI) is a condition often misdiagnosed with early-stage AD typically characterized by subjective memory impairment and modest deficit in main cognitive domains [88]. The advanced stage of AD is associated with severe cognitive decline. Individuals with MCI are memory impaired but have no functional decline and do not meet the requirements for dementia [89]. This early stage of AD could lead to worsening of the clinical diagnosis, prolonging the diagnosis period of AD for treatment procedures. Therefore, identifying biomarkers for this pre-symptomatic stage of AD will be the key to successful treatments. It has been reported that brain glucose uptake is reduced by 9% in patients with MCI, allowing for biomarker identification [88].

(b)Clinically Applied Investigations Related to AD

The diagnosis accuracy of neurodegenerative diseases can be significantly enhanced using RS. Most of the published studies rely on the comparison of Raman features of healthy donors with that of infected patients. Diagnosis of AD is associated with the abnormal formation of amyloid plaques and neurofibrillary tangles. Therefore, many researchers developed Raman-based strategies to detect such proteins aggregated in biofluids and tissue samples. For example, Lochocki et al., employed RS to study Aβ deposition in AD tissue sections from post-mortem patients [90]. Additionally, Ryzhikova et al., suggested that early detection of AD is possible using RS investigation of CSF [91]. In that study, researchers resolved AD diagnosis with 84% sensitivity and specificity. In another study, the same researchers revealed the applicability of blood serum for AD diagnostics [92]. Moreover, Carlomagno et al. used multivariate statistical analysis of human serum for AD disease evaluation [93].

RS is one of the most popular methods for profiling cellular metabolites such as neurotransmitters. Imbalances in neurotransmitters are directly correlated to NDs such as AD and PD. As metabolite monitoring is an important aspect in clinical studies, significant research progress has been made for spatial localization of neurotransmitters in living cells [94]. Manciu et al., revealed the clinical potential of RS for the detection and monitoring of neurotransmitters [95]. The study carried out real-time detection of serotonin, adenosine, and dopamine in vitro. Recently, Fu et al., used an animal model for label-free imaging of acetylcholine using SRS [96]. The authors used vibrational signatures of acetylcholine at 720 cm^−1^ to quantify its local concentration directly at the neuromuscular junctions of the frog cutaneous pectoris muscle. SERS has been used to monitor the concentrations of various neurotransmitters and fibrilization of AD-responsible proteins. SERS measurements have been previously reported for the detection of several neurotransmitters, including dopamine, melatonin, serotonin, GABA, and acetylcholine [97,98,99,100]. These neurotransmitters are particularly important, as they are useful as biomarkers for the diagnosis and monitoring of neurological diseases. More recently, Moody et al., carried out a comprehensive study of SERS detection using seven neurotransmitters to establish optimal detection conditions such as type of metal and wavelength [101]. Additionally, they used PCA analysis to decompose large datasets and to identify spectral patterns. For serotonin, GABA, and glutamate, the best limit of detection (LOD) was achieved with silver nanoparticles as a SERS substrate at an excitation wavelength of 633 nm. On the other hand, for melatonin, dopamine, epinephrine, and norepinephrine, the best LOD was achieved using gold nanoparticles (AuNPs) at an excitation wavelength of 785 nm. In a different study, Ende et al., demonstrated physicochemical trapping of neurotransmitters based on AuNPs and polyvinylpyrrolidone (PVP) to detect molecules that are weakly affinitive to gold [102]. Additionally, Lee et al., reported the spread spectrum SERS (ss-SERS) technique to detect neurotransmitters at the attomolar level, opening opportunities for early diagnosis of neurological disorders [103]. Experimental results showed improvement in the SNR of more than three orders of magnitude. Such an exceptional SNR enhancement allows for the ultrasensitive detection of neurotransmitters.

**Figure 3 biosensors-13-00027-f003:**
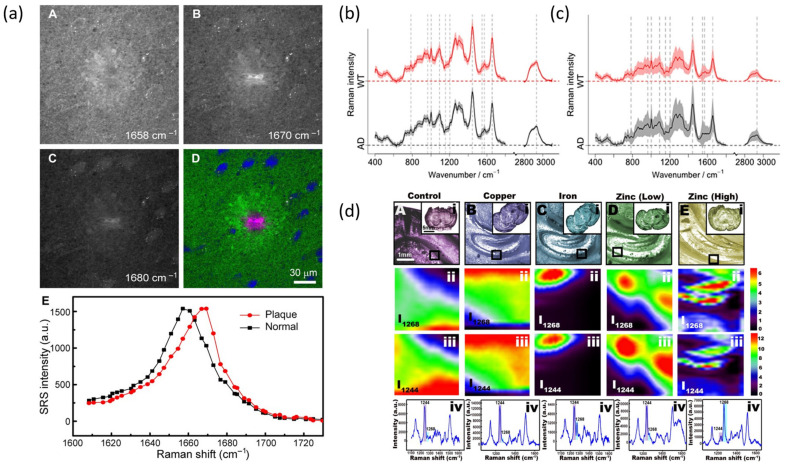
Utility of Raman spectroscopic techniques in diagnosis of AD. (**a**) SRS images of fresh mouse brain sections at (**A**) 1658, (**B**) 1670, and (**C**) 1680 cm^−1^. (**D**) Three color images showing the distribution of lipids (green), proteins (blue) and amyloid plaque (pink) in the mouse brain tissue. (**E**) SRS spectra of the 1600–1720 cm^−1^ region, showing a 10 cm^−1^ shift of the amide I band. Adapted with permission from Reference [65] © American Association for the Advancement of Science. Overall mean spectra based on the two groups, wild-type (WT) (red) and AD mice (black), of the en face Raman measurements used for the (**b**) classification model and (**c**) the cross sections. Adapted with permission from Reference [71] © 2020 American Chemical Society. (**d**) SERS imaging of Aβ_40_ in brain tissues where (**i**) Bright-field of tissue slices from 2-month-old APP/PS1 transgenic mice treated with: (**A**) control diet, (**B**) Cu^2+^, and (**C**) Fe^3+^, Zn^2+^ of (**D**) low and (**E**) high concentration incubated with our SERS platform for 90 min. SERS imaging of Aβ_40_ in hippocampus of tissue slices: (**A**) control diet, (**B**) Cu^2+^, and (**C**) Fe^3+^, Zn^2+^ of (**D**) low and (**E**) high concentration. (**ii**) represents *I*_1268_ and (**iii**) represents *I*_1244_. (**iv**) SERS spectra of Aβ_40_ in hippocampus of tissue slices from 2-month-old APP/PS1 transgenic mice treated with: (**A**) control diet, (**B**) Cu^2+^, and (**C**) Fe^3+^, Zn^2+^ of (**D**) low and (**E**) high concentration. Adapted with permission from Reference [84] © 2020 American Chemical Society.

Clinical evaluation of suspected MCI patients has similarities to that for AD including but not limited to the history of the patient, mental status examination, and medical laboratory tests. Cerebrospinal fluid (CSF) measures of Aβ and tau are useful as biomarkers in predicting the progression from MCI to AD [89]. Hansson et al. revealed the potential of CSF concentrations of Aβ_1–42_, total tau (t-tau), and tau phosphorylated at threonine 181 (p-tau) in predicting the progression from MCI to AD [104]. Cennamo et al. demonstrated SERS of tear fluid as a potential source of a biomarker to differentiate AD- and MCI-affected patients [105]. They used 18 AD-affected and 7 MCI-affected patients including both men and women. Spectral differences were characterized in different regions attributed to lactoferrin and lysozyme protein components. Additionally, researchers used PCA analysis to discriminate AD- and MCI-affected patients. In another study, Raman spectroscopy was used to analyze saliva samples collected from AD and MCI individuals and achieved greater than 99% accuracy [106]. In the end, researchers proved that RS in combination with ML is successful as an accurate diagnostic method in the early stages of AD.

#### 4.1.2. Parkinson’s Disease (PD)

After AD, PD is the second most common ND. Weight loss and behavioral abnormalities are common symptoms of PD. The neuropathological hallmark of PD includes abnormal deposition of a protein called α-synuclein and dopamine deficiency [107]. The native form of α-synuclein is intrinsically disordered, and it undergoes a transition of the structure due to PD [108].

(a)Fundamental Investigations Related to PD

RS has been utilized to characterize the secondary structure of α-synuclein and its aggregation. Raman optical activity (ROA) is a chiroptical spectroscopic technique useful in identifying the secondary structure of proteins. This technique is based on the difference in scattering intensities between left and right circularly polarized light. Mensch et al., used ROA with conventional RS to detect the transition of α-synuclein from a disordered form to α-helical or β-sheet forms [109]. They used increasing concentrations of fluorinated alcohols to induce aggregation of α-synuclein and identified states that act as intermediates for aggregation and β-sheet-rich oligomers. It is of great importance to understand the aggregation process of α-synuclein, as it is an important drug discovery target for PD. Toward that goal, RS was utilized to identify differences in normal and fibril states of α-synuclein. Maiti et al., used a three-component band fitting (α-helix ~1650–1656 cm^−1^, β-sheet ~1664–1670 cm^−1^, and unordered ~1680 cm^−1^) of the amide I region to investigate the secondary structure during aggregation of α-synuclein [110]. The results estimated that 48% of the secondary structure is composed of α-helix. In a follow-up study, the same research group carried out further analysis of the α-synuclein amide I region during fibrillation [111]. Data revealed that the transition of a monomer to aggregate is a complex phenomenon that results from the interplay between various processes (Figure 4b). The results have validated that one of the intermediates a-synuclein aggregation possesses a-helical conformation. Additionally, an increase in β-sheet content and a decrease in the disorder of protein were observed during aggregation.

In particular, the SERS study on α-synuclein is scarce. In an early study, Zhang and coworkers designed a liquid core photonic crystal fiber (LCPCF) and SERS-based sensor for α-synuclein detection [112]. More recently, SERS-based microfluidic testing chips have been constructed for the investigation of the transient species of α-synuclein at physiological concentration [54]. Briefly, optical tweezers were used to tune the separation of two silver nanoparticle-coated silica microbeads, allowing for precise control of the hotspot. The 200 parallel SERS measurements of 1 µM α-synuclein solution are characterized by unique Raman fingerprints attributed to α-helix and β-sheets (Figure 4a). Further, this method allowed for a LOD of α-synuclein as low as 100 nM. Since dopamine deficiency is a neuropathological hallmark of PD, several studies have applied SERS to the detection of dopamine. An et al., used AuNPs immobilized on a glass substrate for the detection of dopamine and obtained a detection limit of 1 nM [55]. The Raman spectrum of dopamine displayed broad bands at 1267, 1331, 1158, 1478, 1578, and 1584 cm^−1^ with peaks at 1267 and 1478 cm^−1^ identified as phenolic carbon–oxygen and phenyl C=C stretches, respectively. The Fe_3_O_4_/Ag nanocomposite has been successfully applied for the determination of dopamine in artificial cerebrospinal fluid and mouse striatum [113]. Many studies have demonstrated SERS as a facial-sensing strategy for dopamine. An in-depth discussion of SERS as a detection method for dopamine is out of the scope of this review, and readers are encouraged to read other articles on this topic [114,115].

(b)Clinically Applied Investigations Related to PD

The strategy of collecting biochemical information at the structural level of biological organization is applicable in other NDs such as PD. SERS has been used to probe the dopamine, human dopamine transporter, and dopamine–human dopamine transporter (DA-hDAT) interactions in live cells [116]. The analysis of experimental results revealed that Raman wavenumbers of 807 and 1076 cm^−1^ are crucial for the DA-hDAT interactions. These peaks are attributed to bound states of dopamine molecules in the human dopamine transporter. Furthermore, analysis of physiological dopamine concentration in complex biological fluids was also reported as an alternative diagnostic test for PD [117]. For example, Phung et al. reported 86% lower dopamine concentration for patients with drug-induced Parkinsonism compared with the level in a healthy human body [56]. The average dopamine concentrations of the two groups were 2.31 × 10^−8^ and 3.24 × 10^−9^ M for healthy and infected samples. Further, the results demonstrated the detection of dopamine concentration as low as 10^−11^ M (Figure 4d,e). These findings highlight the applicability of the SERS technique as an ultrasensitive detection platform to diagnose PD through dopamine, as its concentration in samples is relatively low. In another study, Schipper and coworkers developed an innovative platform for plasma metabolomics for biomarker studies of PD [118]. In that study, the researchers demonstrated that RS and near-IR spectroscopy (NIRS) of plasma differentiate individuals with idiopathic from healthy samples with ~75% accuracy. This finding was based on significant variations in oxidative stress sensitivity bands in comparison with the control. Additionally, the study mentioned that reactive oxidative species (ROS) modify the proteins, lipids, and other cellular substrates in plasma. Carlomagno et al. used the saliva of PD patients to create an automatic classification model [119]. In that study, Raman spectroscopic analysis was applied to the saliva of 23 PD patients and 33 healthy samples. Acquired data were further analyzed using ML techniques. The proposed method highlights the potential to determine PD onset and progression, monitor therapies, and rehabilitation efficiency.

Brain disorders affect the normal functions of the retinal layers and their subsections. Many studies have shown that PD has a direct correlation with visual dysfunctions, including color discrimination, visual activity, contrast sensitivity, blurred image, motion perception, and loss of vision [120]. Mammadova et al., used RS to investigate retinal pathology in a transgenic mouse model (TgM83) expressing the human A53T α-synuclein mutation [121]. In that work, α-synuclein was shown to accumulate in the inner and outer retina of 8-month-old TgM83 transgenic mice, expressing A53T human α-synuclein under the control of the Prnp promoter. Phospho-α-synuclein was only present in the outer nuclear layer. In addition, TgM83 transgenic mics showed increased microglial activation followed by increased GFAP immunoreactivity. Bedoni and coworkers examined saliva from PD, AD, and amyotrophic lateral sclerosis (ALS) patients showing key spectral differences [122]. In this study, RS was used to detect biomarkers of ALS compared not only to controls but also to PD and AD patients (Figure 4c). They mentioned that this approach can drastically shorten diagnosis times that lead to precise and quick diagnoses of the most dangerous neurogenerative diseases. Overall, the above findings suggest that different Raman techniques correlate well with clinical trials offering an easy and user-friendly tool for disease diagnosis.

**Figure 4 biosensors-13-00027-f004:**
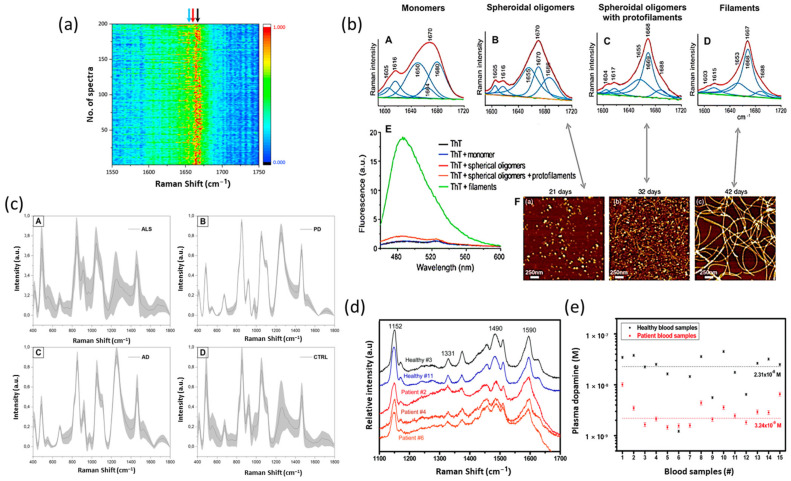
Applicability of different Raman techniques in diagnosis of PD. (**a**) Mapping of 200 SERS spectra of 1 μM alpha-synuclein solution obtained from two AgNP-coated beads trapped at 20 nm with 1 s acquisition time. The color bar shows the normalized intensities from low (dark blue) to high (red). The blue arrow represents the amide I band at around 1653 cm^−1^, the red arrow represents the amide I band at around 1664 cm^−1^, and the black arrow represents the amide I band at around 1671 cm^−1^. Adapted with permission from Reference [54] © 2021 Nature Portfolio. (**b**) Structural changes of α-synuclein during aggregation. Raman spectra (**A**–**D**) reveal a narrowing of the amide I band with an increase in intensity of the peak ∼1670 cm^−1^, indicating an increase in β-sheet structures with aggregation. (**E**) ThT fluorescence spectra only show a large increase in fluorescence on filament addition, despite the high level of β-sheet detected in the protofilament sample by RS. (**F**) Spheroidal oligomers were observed at 21 days of incubation, with protofilaments at 32 days and filaments at 42 days of incubation. Adapted with permission from Reference [111] © 2006 Elsevier. (**c**) Average Raman spectra with SD of (**A**) ALS, (**B**) PD, (**C**) AD and (**D**) control groups. Adapted with permission from Reference [122] © 2020 Nature Portfolio. (**d**) Raman spectra of plasma dopamine extracted from the blood samples of healthy subjects and patients and (**e**) plasma dopamine levels of all blood plasma samples using the SERS technique. Adapted with permission from Reference [56] © 2018 Royal Society of Chemistry.

#### 4.1.3. Huntington’s Disease (HD)

HD is a progressive ND that belongs to the category of autosomal-dominant disorders. This is caused by the expansion of CAG trinucleotides coding for the poly-glutamine (poly-Q) stretch at the NH2-terminus of the huntingtin (Htt) protein. Thus, mutated Htt is the cause of HD. Patients with HD characteristically lose weight and are observed with motor, psychiatric, and cognitive abnormalities. Similar to AD and PD, its pathogenesis is related to the aggregation and accumulation of misfolded proteins in peripheral nerves. HD has no cure, and most of the diagnoses are performed by genetic testing. Therefore, finding suitable biomarkers for the detection and identification of the onset of HD could be beneficial and enable therapeutic intervention.

(a)Fundamental Investigations Related to HD

RS has been used for the quantification and visualization of aggregated proteins and other aspects of HD. Miao et al., reported a novel platform for live-cell imaging of aggregates by combining SRS microscopy with Gln-d5 labeling [43]. This combined approach facilitated measuring absolute concentrations of sequestered mutant Htt and other proteins within the same aggregate. It has been demonstrated more recently that UV resonance Raman spectroscopy (UVRRS) is useful to monitor polyglutamine backbone, side chain hydrogen bonding, and fibrillization [44]. CARS microscopy has been used to image polyglutamine aggregate structures in vitro and in vivo [123].

(b)Clinically Applied Investigations Related to HD

SERS and spontaneous RS of serum were used to identify disease progression of HD [20]. The study mentioned that there are significant differences corresponding to genotype and gender in serum samples of HD patients and healthy controls. For HD patients, Raman bands at 1245 and 1667 cm^−1^ are dominant, indicating higher content of β-sheet protein structures present in the HD serum compared to healthy controls (Figure 5). Peripheral fibroblasts are useful as a potential model for HD. In this regard, RS has also been used to identify living fibroblast (skin) cells from an HD patient with an accuracy of 95% [124]. Raman spectrum from HD patients revealed that more β-sheet proteins are present at 1220 cm^−1^. This is consistent with the finding that the aggregation of the protein from the mutated huntingtin gen, is known to take a beta-sheet form. Additionally, there are other differences in HD patients, including reduction in the amount of lipids and cholesterol.

In another study, RS and partial least squares analysis was used in the discrimination of peripheral cells affected by HD [45]. Significant differences were observed in the low wavenumber region (400 to 1800 cm^−1^). HD patients showed differences in the Raman peaks at 428 and 701 cm^−1^, which is indicative of cholesterol and cholesterol esters, in comparison to control samples. Additionally, differences were seen at 1045, 1073, and 1130 cm^−1^ regions corresponding to triglycerides, phospholipids, fatty acids, and proteins. For healthy samples, three peaks are prominent (548, 1331, and 1685 cm^−1^), which were missing in HD patients. Those peaks were attributed to cholesterol, phospholipids, and proteins. Overall, this study is an excellent example demonstrating biological fluids as useful biomarkers for HD diagnosis. All these findings clearly illustrate that Raman-based techniques are excellent tools in the early diagnosis of HD paving a path toward clinical translation.

### 4.2. Brain Tumors

Brain tumors account for 90% of all central nervous system tumors and occur due to the growth of malignant cells in tissues of the brain. Advances have been made using different techniques such as PET, ultrasound (US), MRI, and optical coherence tomography (OCT) to provide structural information and surgical planning of brain tumors [125,126,127]. Such conventional methods have limitations, including the inability to capture tumor heterogeneity, low sensitivity, and low resolution. Therefore, there is an urgent need to develop new methods that can detect tumor presence non-invasively. Over the past couple of decades, research efforts have leveraged the benefits of Raman techniques for brain tumor diagnosis. The brain contains many Raman active species, allowing it to be an efficient and non-invasive technique in brain cancer detection. In addition, RS strategies for label-free spectroscopic analysis of brain tumors have allowed for accurate diagnosis of brain tumors and assessment of surgical outcomes.

(a)Fundamental Investigations Related to Brain Tumor

Detection of brain cancer-specific biomarkers in the blood is somewhat challenging due to the limited exchange of biomolecules. The development of SERS-based nanosensors has recently been shown to address this issue, which enables deep brain cancer surveillance [128] (Figure 6a). More recently, Premachandran et al., reported a Ni-NiO-based SERS platform for the detection of blood-based molecules that helps to accurately detect the presence of primary and secondary tumors [129]. The developed hybrid SERS substrate helps to combine electromagnetic enhancement from metallic Ni as well as chemical enhancement due to the charge transfer mechanism. Detection is based on Raman molecular profiles of sera with a minimal working volume of 5 µL. Raman spectrum of brain cancer revealed signature peaks assigned to lipids, fatty acids, and proteins. The specificity of the developed platform for cancer detection was further revealed by comparing molecular fingerprints of brain cancer sera with that of breast, lung, and colorectal cancers (Figure 6d,e). Additionally, the developed method could identify the exact tumor location based on species such as glycogen, phosphatidylinositol, nucleic acids, and lipids. Kircher et al. reported a combination of SERS, PA, and MRI to visualize brain tumor margins with high precision using Au nanotags functionalized with Gd organometallic complexes [130]. This approach of combining endoscopic, photoacoustic, and Raman imaging capabilities would open a possibility of clinical translation of the MPR approach (magnetic resonance imaging–photoacoustic imaging–Raman imaging nanoparticle). Li and coworkers developed a surface-enhanced resonance Raman scattering (SERRS) probe using gold nanostars and IR-783 dye [131]. The developed SERRS probe demonstrated an ultrahigh detection limit of 5 pM in an aqueous solution.

Prasad and coworkers utilized CARS to monitor the intense upregulation of protein and lipid synthesis signals in microglia cells [132]. Their results demonstrate the activation of microglia in the presence of bacterial liposaccharide due to the action of proteins and lipids, further verifying the potential of CARS in the detection of neurological diseases. Koljenovic et al. showed that fiber-optic Raman probes used to collect Raman-scattered light in the high wavenumber spectral region (2400–3800 cm^−1^) can be used to characterize porcine brain tissue ex vivo [133]. The authors evaluated coronal plain sections of seven pig brains. Based on the biochemical differences revealed by Raman spectra, they were able to distinguish adjacent brain structures. In a different study, this group examined 20 unfixed cryosections of glioblastoma by RS for separating vital and necrotic tissues [134]. Spectral signatures resembled that of cholesterol and cholesterol esters consistent with the increased presence of cholesterol in necrotic tissues. Cluster analysis revealed 100% diagnostic accuracy.

(b)Clinically Applied Investigations Related to Brain Tumor

Several Raman techniques were investigated in guiding brain tumor diagnosis. For instance, spontaneous Raman scattering could be useful in identifying tumor margins, tumor infiltration zones, brain edema, and tumor recurrence [130,135,136,137]. Jermyn et al., reported the use of RS for the intraoperative detection of brain cancer in a clinical trial of humans [138]. Their hand-held contact RS probe technique could distinguish a normal brain from dense cancer and a normal brain invaded by cancer cells, with a sensitivity of 93% and a specificity of 91% (Figure 6b,c). This RS system’s success in clinical utility was enabled by an optical probe to maximize photon collection efficiency. Minimizing the volume of residual cancer is an important factor in clinical practices. This study estimated the cellular resolution of the Raman probe, with the detection of as few as 17 cancer cells/0.0625 mm^2^, further verifying the utility of this technique in rapid cancer detection. The presence of cancer cells was detected using lipid bands (700 to 1142 cm^−1^), nucleic acid bands (1540 to 1645 cm^−1^), and the phenylalanine band in proteins (1005 cm^−1^). With subsequent studies, researchers were able to commercialize the optical probe for clinical translation [139,140,141]. A similar probe was reported that combines RS, intrinsic fluorescence spectroscopy, and diffuse reflectance spectroscopy that is translatable to the diagnosis of other cancers [142]. Recently, an imaging needle was developed for intraoperative detection of blood vessels during neurosurgery in humans [143]. In another study, Kircher and coworkers demonstrated the potential of SERS and optoacoustic tomography for intraoperative brain tumor delineation, thereby improving surgical care [144]. The authors mentioned that the proposed dual-modal concept is suitable for clinical translation due to the acceptable illumination energy used throughout the experiment. Using SERS, guiding brain tumor resection is also possible. The breakthrough demonstration reported by Kircher’s group showed instrumentation that can aid in brain tumor resection [145]. They used a hand-held Raman scanner to target glioblastoma tissues intraoperatively in genetically engineered mouse animal models. In another work, Hollon et al., demonstrated stimulated Raman histology as a powerful technique for near real-time intraoperative brain tumor diagnosis [146]. By combining convolutional neural network (CNN) with stimulated Raman histology, researchers were able to achieve 100% classification accuracy. By leveraging recent developments in deep learning to train CNN on more than 2.5 million SRH images, researchers were able to predict brain tumor diagnosis in the operating room in under 150 s, which is significantly faster than conventional techniques. The outcome of this clinical trial demonstrates how stimulated Raman histology as a complementary pathway for tissue diagnosis can improve the care of brain tumor patients. Overall, the above-mentioned findings create the possibility of translating Raman-based techniques from the laboratory to the clinic.

Recent progress with the use of CARS for discerning healthy cells from tumor cells is highly promising. Galli et al., used a combination of CARS, two-photon excited fluorescence, and green fluorescence protein (GFP) labeling to identify glioblastoma tumors and infiltrates [137]. In this study, human tissue samples were collected during brain surgeries. The cell morphology and chemical contrast provided by CARS allowed for tumor recognition and localization of infiltrating tumor cells. Uckermann et al., employed CARS for the detection of different human brain tumors in a mouse model [147]. Here, C-H molecular vibration was used as a probe to distinguish the lipid content of the sample since all brain tumors have significantly low lipid content (Figure 6f). SRS has shown the capacity to reveal features of tumor tissues similar to the standard H&E stain method [148]. Camelo-Piragua and coworkers demonstrate an SRS-based technique in a clinical operating room to improve the surgical care of brain tumor patients [149]. Here, they developed a portable, fiber-laser-based SRS microscopy system for rapid intraoperative tissue processing. Additionally, a clinical SRS microscope has been designed and utilized in operating rooms [150]. In this report, researchers developed a method based on stimulated Raman histology to avoid time-, labor-, and resource-intensive standard H&E histology. This method was able to produce 2 × 2 mm SRH images at the bed site within 90–120 s.

**Figure 6 biosensors-13-00027-f006:**
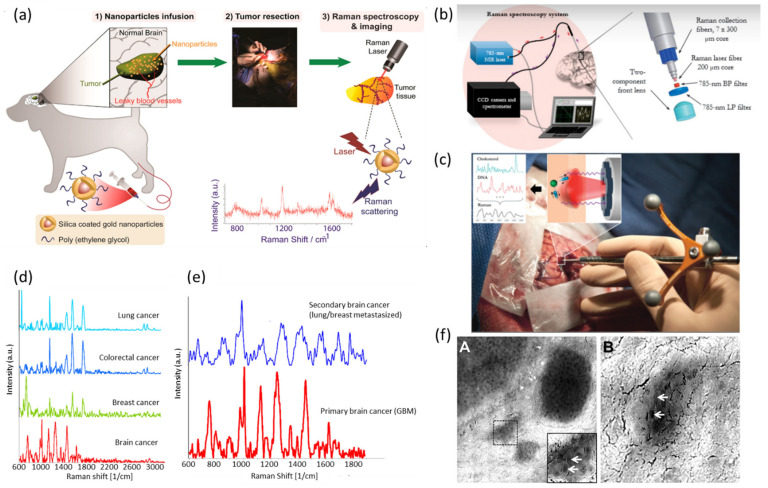
Illustration of applicability of Raman techniques in diagnosis of brain tumors. (**a**) Schematic showing the overall design of the experiments starting from nanoparticle preparation to intravenous infusion, surgical resection, and analyses. Adapted with permission from Reference [128] © 2019 American Chemical Society. (**b**) Experimental setup diagram with the 785 nm NIR laser and the high-resolution CCD spectroscopic detector used with the Raman fiber optic probe. (**c**) The probe (Emvision, LLC) used to interrogate brain tissue during surgery. Inset shows the excitation of different molecular species, such as cholesterol and DNA, to produce the Raman spectra of cancer versus normal brain tissue. Adapted with permission from Reference [138] © 2015 Science. (**d**) SERS spectra of sera from brain, breast, lung, and colorectal cancer. (**e**) Raman spectral profiles of serum of brain cancer patients and serum of metastasized brain cancer. Adapted with permission from Reference [129] © 2022 American Chemical Society. (**f**) (**A**) CARS image of a human U87MG glioblastoma in a mouse brain and (**B**) CARS image of a separate small glioblastoma island in a mouse brain. Single cell nuclei appear as dark structures in the tumor denoted by arrows. Adapted with permission from Reference [147] © 2014 Public Library of Science.

A recent study by Bury et al., analyzed 29 brain tissue samples that had been obtained during surgery [27]. Using gold nanoparticles as a SERS substrate and a handheld Raman device, researchers were able to differentiate tumor types from fresh brain tissue. Another clinically relevant study by Ji et al., used SRS to identify human brain tumor infiltration in surgical specimens from 22 neurosurgical patients [151]. By constructing two-color images based on Raman intensity ratios, they were able to identify structures that were lipid or protein rich. They reported a sensitivity of 97.5% and a specificity of 98.5% of detecting tumor infiltration. In an alternative approach, Desroches et al., developed an instrument using a core needle biopsy probe for detecting dense human brain tumor [26]. This instrument can be used in situ during surgery and has minimal impact on the flow of clinical procedure. Using high wavenumber Raman spectroscopy, cancer cells were detected with 80% sensitivity and 90% specificity. In a follow-up work, the same researchers developed a navigation-guided fiber optic Raman probe that allows surgeons to interrogate brain tissue in situ at the tip of the biopsy needle prior to tissue removal [152]. Feature engineering was used to develop a new representation for spectral data tailored for brain tissue diagnosis in a clinical setting [21]. This method was based on a dataset of 547 in vivo Raman spectra of 65 patients. In contrast to conventional imaging techniques used for tumor diagnosis, the spectroscopic signatures provided by Raman techniques provide additional information about molecular information regarding tissues and cell-to-cell heterogeneity. Raman mapping in combination with the PLS method was used to predict the tumor amount in dura and meningioma obtained from 20 patients during a neurosurgical procedure [153]. Raman spectra of dura is characterized by higher collagen content while lipid content of meningioma is significantly higher. Results of this work opened an avenue for the development of an in vivo Raman spectroscopy method for real-time guidance of meningioma resection. Leblond and coworkers reported optimum conditions of a Raman spectroscopy setup suitable for neurosurgery [139]. They demonstrated that SNR increased as the camera temperature decreased and integration time increased. Additionally, they revealed that external sources of light such as a microscope light, operating room lights, LCD screens, and daylight leakage impaired the ability of accurate Raman measurements of the sample. Overall, different Raman techniques are now steadily becoming popular and applicable in the clinical diagnosis of brain disorders.

## 5. Conclusions and Future Prospects

In the past few years, with the advances in spectroscopic tools and nanoscience, the diagnosis of brain disorders has made great progress. RS can assist in uncovering pathways of brain disorder progression. Several studies have demonstrated the capability of RS for identifying tissue classification of different areas of the brain as well as identifying different variants of brain pathologies. The applicability of Raman spectroscopic techniques in the diagnosis of brain disorders continuously expands due to their effectiveness. The current interest of researchers is to establish a place for RS in standard clinical practices. The translation of RS toward clinics has been amplified due to technological advancements alongside continued research breakthroughs that highlight clinical applications.

Despite the remarkable work presented in this review in the field of RS in clinical applications of brain disorders, there remain several challenges that stand in the way of clinical transition. It has been known to the scientific community that RS has suffered from drawbacks such as weak signals, long acquisition times, fluorescence from biological samples, time-consuming data processing, and costs. Remarkable progress has been made over the past decades to address these challenges with the help of advancements in instruments and ML techniques. To enhance weak signals and improve SNR, several complementary techniques such as SERS, RRS, and SRS are useful. Additionally, instrumental design is heading toward gathering data with high resolution, high accuracy, and fast acquisition times. The consistency of the sample measurements is pivotal for the transition from benchtop to bedside. It involves the establishment of profound spectral databases and the need for inter-system calibration. Calibration should be performed using National Institute of Standards and Technology (NIST)-approved reference materials. Additionally, one of the translational hurdles involves the variable results from different Raman setups. Therefore, attention should be focused on defining methodologies and developing ML models and chemometric methods to account for undesirable variations. The roadmap of the translation of RS into clinical studies also involves clinical trials, regulatory approval, FDA guidelines, and market assessment. The key to obtaining sufficient data and their interpretation is based on suitable animal models to study biomarker identification and disease progression. Sometimes it could be challenging to measure the Raman signal in the presence of extraneous light sources. Therefore, engineering solutions based on proper light filtering can be used to minimize this effect in clinical settings. Overall, close collaborations between spectroscopists, material scientists, biomedical engineers, and clinicians are required to make the clinical transformation of RS into a reality.

There are still challenges in deploying ML methods in practical clinical diagnosis. One of the challenges is to select a proper model. The Raman spectra are high-dimensional and with noise. Therefore, the models can be easily overfitted with overcomplex ML models. To achieve high accuracy in classification and to prevent overfitting, the complexity of the model must be carefully selected with experts and cross-validation techniques. In addition, the ML frameworks should be modified specifically for Raman spectra to recognize the patterns and correlation of Raman peaks. Another challenge of ML methods is interpretability. Many complex ML models can achieve high performance in classification, while the interpretation of these models is hard. The ML methods act as black boxes and cannot understand the problem. However, in clinical diagnosis, the robustness and interpretation of diagnosis are critical. The lack of transparency in classification and diagnosis limits the practical deployment of the technique. To resolve the interpretability of ML methods, linear models are preferred since they are easy to explain. Some feature selection techniques can be potentially extended and applied to rationalize the decision-making process in analyzing clinical Raman spectra in brain diseases and cancers.

We anticipate the future development of RS in clinical trials of NDs on several fronts. At present, 2D materials are rarely used in conjunction with Raman techniques for biomarker detection and disease progression on brain disorders. Therefore, exploring various disciplines of 2D material-assisted RS is an effective approach for future directions. On the way of moving forward with 2D material-assisted RS, several factors need to be carefully researched, such as material performances, stability in a biological medium, large-scale production, and biosafety. In this regard, surface functionalization of 2D materials that improve biocompatibility and colloidal stability needs to be thoroughly investigated. Research also needs to focus on integrating RS with other spectroscopic techniques to design multimodal techniques that can provide additional and complementary information on clinical settings. Spectroscopic identification can be somewhat challenging when multiple analytes are present in complex biofluids. To overcome this issue, Raman techniques can be hyphenated with separation techniques such as liquid chromatography. Additionally, the focus should be aimed at the simultaneous detection of multiple biomarkers. Future Raman-based devices should be automated as much as possible to minimize the burden on the clinical community. In the future, ML–Raman techniques may further improve the accuracy and reduce the time and cost in the early diagnosis of various brain diseases and cancers. With the ability to analyze a large number of spectra and recognize the pattern, ML technologies can also be further developed to rapidly identify biomarkers and, therefore, facilitate drug development. The availability of open Raman datasets, open-source libraries, and high-performance computing resources will also accelerate the progress in applying different existing ML methods and developing new ML algorithms in analyzing clinical Raman spectra. We envision that the future of precision medicine in clinics will be based on robotics. Therefore, necessary steps should be taken to design Raman-based techniques with robots. Overall, the rapid development of Raman-based techniques and ML capabilities is continuously pushing the boundaries in clinics to improve patients’ well-being. We hope this review will open a new avenue to this burgeoning field.

## Figures and Tables

**Figure 1 biosensors-13-00027-f001:**
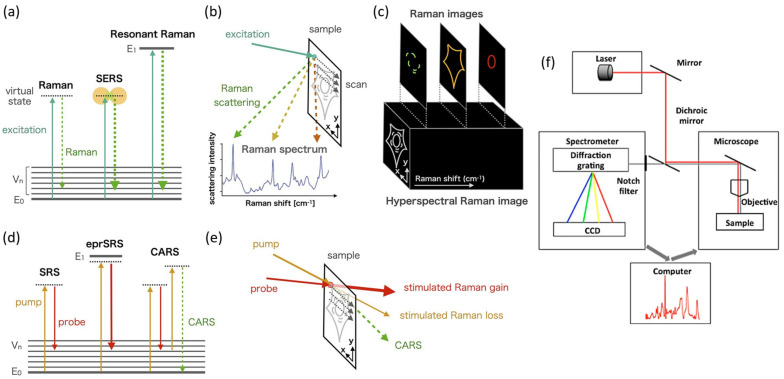
Principles of RS: (**a**) energy level diagram showing Raman scattering, SERS, and RRS. E_0_, E_1_, and V_n_ show the electronic ground state, an electronic excited state, and vibrational excited states, respectively. (**b**) Raman spectrum induced by laser light focused on a sample during Raman microscopy. (**c**) Spatial distribution of Raman spectra, also referred to as hyperspectral Raman images, where Raman images are obtained as distributions of Raman peak intensities. (**d**) Energy level diagram of SRS, electronic pre-resonant stimulated Raman scattering (eprSRS), and CARS. (**e**) SRS microscopy detects the energy exchange between the pump and probe beams via the vibrational excitation state as stimulated Raman gain (probe beam) or loss (pump beam) to reconstruct a Raman image. CARS microscopy uses CARS signals emitted from the sample as the image contrast. Adapted with permission from Reference [11] © 2021 American Chemical Society. (**f**) Generic setup for a Raman microspectroscopy system. Adapted with permission from Reference [12] © 2018 American Chemical Society.

**Figure 5 biosensors-13-00027-f005:**
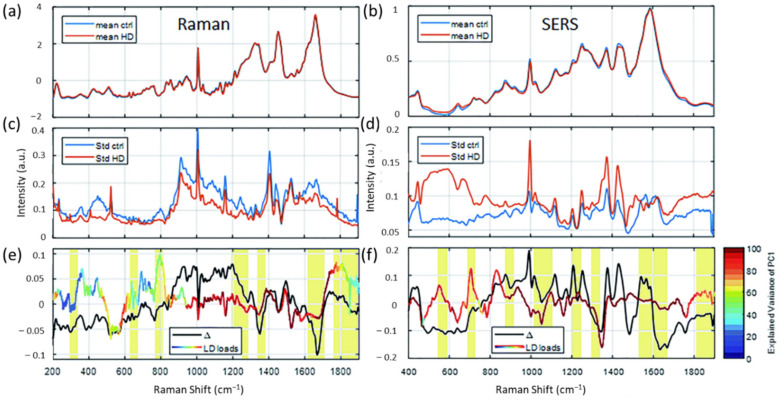
Illustration of applicability of Raman techniques in diagnosis of HD. Average RS (**a**) and SERS (**b**) spectra of serum from healthy control subjects (blue lines) and HD patients (red line) as well as their standard deviations ((**c**,**d**), respectively). The different spectra of the averages for RS (black line, (**e**)) and SERS (black line, (**f**)) are within the standard deviation (**c**,**d**) of the average spectra (**a**,**b**). Yellow marked regions indicate important peaks. Adapted with permission from Reference [20] © 2020 Royal Society of Chemistry.

**Table 1 biosensors-13-00027-t001:** Mechanisms, biomarkers, Raman sensitivity, and diagnose methods other than Raman spectroscopy for NDs.

	AD	PD	HD
Mechanism	Aβ Protein misfolding [10] Hyperphosphorylation of tau causing aggregation [10,37]	Aggregation of α-synuclein [10,37]	Expansion of CAG trinucleotides coding for poly-glutamine (poly-Q) stretch at the NH_2_-terminus of the huntingtin (Htt) protein [10,37]
Biomarkers	Tau proteins (t-tau, p-tau) [38] Aβ (Aβ oligomer, Aβ40, Aβ42) [38] Neurofilament light chain (NfL) Vinisin-like protein 1 (VLP-1) Neuron-specific enolase (NSE) Heart fatty acid binding protein (HFABP) Glial activation (YKL-40) [6]	α-synuclein [39] Dopamine [39] Orexin [40,41] 8-Hydroxy-2′-Deoxyguanosine [40] miRNA [42]	Hungtintin protein Mutant Htt (mHtt) [43] Polyglutamine [44] Triglycerides, phospholipids, Fatty acids [45] Myelin basic protein (MBP) [46,47] Total tau (t-tau) [48] Melatonin [49] Cortosol [50,51]
Raman Sensitivity	100 fg/mL for Aβ [52] 10^−9^ M for Dopamine [53]	100 nM for α-synuclein [54] 10^−11^ M for Dopamine [55] 1 nM for Dopamine [56]	29 µM for mHtt protein [43]
Diagnose methods other than Raman Spectroscopy	Mental state examination Neurological assessment Brain imaging techniques [6,37,57]	Mental state examination Neurological assessment Brain imaging techniques [6,37,57]	Mental state examination Neurological assessment Brain imaging techniques Genetic testing [6,37,57]

## Data Availability

Not applicable.

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
