# Peer review of "Raman Spectroscopy on Brain Disorders: Transition from Fundamental Research to Clinical Applications"

_biosensors, 2022, doi:10.3390/bios13010027_

Round 1

Reviewer 1 Report

In this review, authors discussed recent clinical and biomedical applications of Raman spectroscopy and related techniques applicable to brain tumors and NDs. The research field of this review is of high impact. The authors have explained the basic knowledge in this field and were familiar with and have highlighted the key points when they were summarizing the previous investigations. This paper has a good organizational framework, and the logic between each chapter or paragraph is clear. So, I think this paper can be recommended for publication after minor revisions.

Here are some suggestions:

1. All the figures are very blurry, and their image resolution does not meet the requirements.

2. The fonts and formats of the figures are inconsistent, and need to be checked.

3. In the chapter introducing machine learning, authors mainly introduced PCA and SVM. Are there other algorithms also widely used? Authors need to explain and add more.

4. In the field of Raman spectroscopy, the signal-to-noise ratio is very important. Recently, some algorithms, such as variational autoencoder (VAE), have been proposed improve the signal-to-noise ratio. Please add some representative references, such as (1) IEEE Transactions on Geoscience and Remote Sensing, 2018, 57(2): 1205-1218; (2) ACS Omega 2022, 7, 12, 10458–10468.

5. Authors introduced the principles and related technologies of PRS, SERS, CARS, and SRS. However, the literature on CARS and SRS is scarce. Can the authors explain it?

Reviewer 2 Report

In the present manuscript, the authors reported numerous examples of the application of spectroscopic techniques based on the Raman effect for the investigation of main brain pathologies. The addressed topic is timely and interesting, but the presentation is confused and rushed. The title is ambiguous since the paper mainly reports studies of basic knowledge on molecular mechanisms underlying the different diseases and some aspects of anatomy imaging, mainly related to animal tissues. Very few true clinical applications aimed at giving an early diagnosis of brain disorder diseases are reported. It is really difficult to understand what is the main focus of the present review and to evaluate in a critical way the real potentialities of Raman-related spectroscopies in the diagnosis of brain pathologies. For these reasons, the paper should be rejected.

Reviewer 3 Report

This review manuscript presents a comprehensive view on applicating Raman spectral technique and machine learning  in  detecting clinical bain disorder  including  Alzheimer's disease (AD), Parkinson's disease (PD), Huntington’s disease (HD), and brain tumors which  represent public health challenges as they can have a profound and even debilitating impact on a patient’s.  This review  will must facilitate the Raman technique  transform toward to clinic application.

Only suggestion is  to individually discuss  clinical  applied investigation  and  basic laboratory research relatived the brain diseases,   thus it is  clearly understood. 

Round 2

Reviewer 2 Report

After the revision, the manuscript is now better structured and the scope of the review clearer, allowing to overcome the ambiguities previously occurring between diagnostic purposes and histological analysis studies. The new title also helps to better connote the purpose of the work. In this form, the manuscript is suitable to be published in Biosensors journal after some minor corrections:

on Page 2, line 10: define the "ML" acronyme.

Figure 3 caption: define again the “wt” acronyme.

Please check the reference list. Some author names have been omitted in the cases where the author number was larger than 10 ( see Ref. 105, 121,122, and 149, for instance). Please, add the missing names or the abbreviation “et al”.
